# Experimental and Numerical Comparison of Impact Behavior between Thermoplastic and Thermoset Composite for Wind Turbine Blades

**DOI:** 10.3390/ma14216377

**Published:** 2021-10-25

**Authors:** Thiago Henrique Lara Pinto, Waseem Gul, Libardo Andrés González Torres, Carlos Alberto Cimini, Sung Kyu Ha

**Affiliations:** 1Structural Engineering Department (DEEs), Federal University of Minas Gerais, UFMG, Av. Antônio Carlos 6627, Belo Horizonte 31270-901, MG, Brazil; thiago.lara@ict.ufvjm.edu.br (T.H.L.P.); cimini@ufmg.br (C.A.C.J.); 2Science and Technology Institute (ICT), Federal University of Jequitinhonha and Mucuri Valleys (UFVJM), Rodovia MGT 367, 5000, Alto da Jacuba, Diamantina 39100-000, MG, Brazil; l.gonzales@ict.ufvjm.edu.br; 3Mechanical Engineering Department, Hanyang University, 222 Wangsimni-ro, Sageun-dong, Seongdong-Gu, Seoul 04763, Korea; mech487@gmail.com

**Keywords:** wind turbine blades, elium thermoplastic matrix, impact, hybrid composites, NCF (non-crimp fabric), VA-RTM (vacuum assisted resin transfer molding)

## Abstract

Damage generated due to low velocity impact in composite plates was evaluated focusing on the design and structural integrity of wind turbine blades. Impact properties of composite plates manufactured with thermoplastic and thermoset resins for different energy levels were measured and compared. Specimens were fabricated using VARTM (vacuum assisted resin transfer molding), using both matrix systems in conjunction with carbon, glass and carbon/glass hybrid fibers in the NCF (non-crimp fabric) architecture. Resin systems used were ELIUM 188O (thermoplastic) from Arkema Co., Ltd. and a standard epoxy reference, EPR-L20 from Hexion Co., Ltd. (thermoset). Auxiliary numerical finite element analyses were performed to better understand the tests physics. These models were then compared with the experimental results to verify their predictive capacity, given the intrinsic limitations due to their simplicity. Based in the presented results, it is possible to observe that ELIUM is capable to replace a conventional thermoset matrix. The thermoplastic panels presented similar results compared to its thermoset counterparts, with even a trend of less impact damage. Additionally, for both thermoplastic and thermoset resin systems, glass layups showed the lowest levels of damage while carbon panels presented the highest damage levels. Hybrid laminates can be applied as a compromise solution.

## 1. Introduction

Wind is an alternative, renewable and clean energy source with low environmental impact, permanently and globally available [1]. To capture and convert wind energy, aerogenerator turbine blades develop an imbalance between lift and drag forces [2]. The wind and gravity cyclic loading histories that exist at the various locations in the blades suggest that it could be advantageous to use different materials for different parts of the blade in order to avoid failures [3,4,5]. On top of that, the complex blade geometry indicates that advanced composite materials are the best design choice, providing high stiffness-to-weight and strength-to-weight ratios, and adequate fatigue and corrosion resistance, with controlled costs of production [6,7,8,9].

Composite wind turbine blades are exposed to impacts that may occur in the aerogenerator assembling process [10,11,12] and/or operation (hail, birds, etc.) [13,14], leading to the deterioration of structural integrity and load-bearing ability. Induced damage in the form of matrix cracking, delamination, and fiber fracture may threaten the fatigue life of the component [15]. The structural stiffness degradation should be investigated for several types of impact loading, ranging from low velocity impact, which leads to barely visible impact damage with significant effects on compressive strength, to ballistic high-velocity impact, focusing on the residual velocity of the projectile and target energy absorption related to the impact velocity and angle. Usually, computer model simulations of these problems should be performed together with experimental test programs to corroborate the predictions and build confidence. However, wind turbine blades’ full-scale testing is limited due to cost restraints. Therefore, for low velocity impact tests (which are more common for such component), a small-scale coupon test program is more suitable to provide the necessary information. In these tests, a flat, rectangular composite coupon plate is subjected to an out-of-plane, concentrated impact load using a drop-weight device with a hemispherical impactor. The drop-weight potential energy is defined by the specified mass and drop height of the impactor [16,17,18,19]. Information obtained from computer models and experimental results on the coupon-scale analysis can be then used to investigate the behavior of the full-scale component (Figure 1).

Wind turbine blades follow three possible routes after being put out of operation: landfill, incineration or recycling. The two first obviously do not comply with the increasing restrictions for sustainability. Landfill bans for plastics are growing around the world and the incineration byproduct (around 60% scrap as ashes) may contain pollutants and also need to be discarded in landfills [20]. Thus, the material reusability, though limited, partially answers the recent interest in the use of thermoplastic resins for wind turbine blade manufacturing. These thermoplastic resins, developed to replace thermoset resins with equivalent mechanical properties [21,22,23,24], can reduce the component’s carbon footprint during its production and life if material reuse is taken into account. So far, wind turbine blades are mostly manufactured using thermoset composites, which need to be either broken to recycle the remaining fibers (by pyrolysis, solvolysis and other processes) [20,25,26,27,28,29,30] or grinded in order to become feedstock for other applications [20,28,31]. Neither is a complete recycling approach and, therefore, the search for a desirable full material recyclability in a sustainable energy scenario is the target to the development of advanced resins. Corroborating the whole point of lower environmental impact, interest in thermoplastic resins, such as Elium (Arkema Co., Ltd., Lacq, Pyrénées-Atlantiques, France), is increasing due its better reuse capabilities. Moreover, thermoplastic composites combine high impact and damage tolerance properties and good deformation capabilities [32,33,34,35,36,37,38] to efficient and favorable processing and manufacturing [39,40,41]. Thus, though not fully recyclable, a more environmentally friendly thermoplastic matrix can be an attractive alternative to thermosetting matrices, if considering mechanical performance and reuse possibilities.

Although carbon fiber composites are lighter and stronger, wind turbines blades are commonly built of glass fiber composites, mainly due to carbon fiber limited flexibility and higher costs [42]. With the increasing in size of wind turbine blades, hybrid carbon/glass fiber composites became a feasible alternative to balance cost-performance [3,43,44]. They were used to enhance loading carrying and impact resistance, allowing designers to meet the desired performance with marginal cost increase [45,46,47]. The fiber reinforcement hybridization may then be used to achieve the adequate combination of mechanical properties and cost, resulting in high-performance structures that are tougher, stiffer, lighter and economical [48,49,50].

Non-crimp fabric (NCF) composites are very promising materials, combining mechanical properties close to those of unidirectional laminates, with easy and cost-effective production methods [51,52]. It is important to notice that, besides avoiding undulated yarns [53], NCF composites mechanical performance cannot be predicted based on the properties measured on individual layers only, since fibers are assembled in bundles, with a certain degree of undulation, and separated by resin rich areas. This meso-structure will affect the stiffness and strength of the final composite [54], and actions such as systematically removing the stitching from quasi-unidirectional NCFs, before impregnation would be highly impractical [42].

The purpose of this investigation is to compare the impact performance between thermoplastic and thermoset resin systems reinforced by different types of fibers for wind turbine blade design. Composite coupon-sized panels using thermoplastic (ELIUM 188O from Arkema Co., Ltd., Lacq, Pyrénées-Atlantiques, France) and thermoset (EPR-L20 from Hexion Co., Ltd., Colombus, OH, USA) resins, reinforced with carbon, glass and carbon/glass hybrid NCF fabrics were manufactured and tested under impact conditions. Different impact energy levels were applied to the panels and experimental results were compared in terms of absorbed energy and related damage. Different auxiliary numerical model predictions were compared to the experimental results to critically analyze the benefits and drawbacks for each studied case. The results of this investigation can further support impact analysis of full-scale wind energy blades, using the same material systems and impact energies.

## 2. Materials and Methods

The drop weight impact test [55] is an experiment designed to measure impact properties by applying a concentrated load to a composite specimen using a hemispherical impactor drop device. In the presented experiments, the impact properties of thermoplastic (from now on called “Elium”) and thermoset (from now on called “Epoxy”) composite plates were measured and compared for eight different specimen configurations, for different impact energies. Table 1 presents a brief description of each specimen composition and the selected layup sequence chosen for the present studies. These non-symmetric layup sequences were arbitrarily chosen with the purpose of characterizing thermoplastic vs. thermoset, focusing in understanding the material’s behavior. 

Considering the reinforcements, carbon and glass plates were 2.8 mm thick, and hybrid plates, which mixed carbon and glass, were 3.0 mm thick. The NCF architecture considered 0°/90° two-ply arrangement. In the plate Hybrid1, the carbon 0°/90° was laminated on the top face (last ply layered during the manufacturing process) and the glass 45°/−45° was in the bottom face (the first to be layered in the manufacturing process). For the plate Hybrid 2, glass 0°/90° was on the top and carbon 45°/−45° at the bottom. This sequence is particularly important once the impact contact was on the top surface. Figure 2 shows the proposed stacking sequences for specimens (quasi-isotropic lay-ups).

The impact tests were performed using the drop test impact machine, produced by ITOH SEIKI Co. Ltd., Chonburi, Thailand, shown in Figure 3. It consists of a guided impactor system with a total mass of 25 kg, which can be adjusted to initial heights up to 2500 mm; a specimen clamp system; and a load cell, placed inside the impactor, between the impactor base mass (19.4 kg) and the impactor probe (5.6 kg). During the test, the load cell signal is amplified and read by a standard oscilloscope, with voltage output readings directly scaled to force measurements. Additionally, the impact and rebound speeds were obtained by recording each impact process with a high-speed camera.

The experiments were conducted considering three different initial heights of 300, 400, and 500 mm corresponding to three different energy levels, respectively 57, 83 and 97 J. It is important to mention that the impact energy was determined using the impact velocity (kinetic energy) instead of the height (potential energy), since part of the potential energy is dissipated due to friction during the impact procedure.

### 2.1. Manufacturing Process

The thermoplastic resin ELIUM 188O (Arkema Co., Ltd., Colombes, France) was selected to be compared to a standard thermoset epoxy reference, EPR-L20 (Hexion Co., Ltd., Columbus, OH, USA). Both Elium (thermoplastic) and Epoxy (thermoset) resins were applied in conjunction with carbon, glass and hybrid layups to produce the impact test specimens. The selected NCF fabrics are shown in Figure 4.

The sample production process is illustrated in Figure 5. First, a dam is developed with sealant tape on an aluminum tool plate. The dry fabric is then arranged in the respective layup for each laminate case, followed by peel ply, flow media, and two spiral tubes used as resin source and drain. The pack is then covered by the bagging film. Vacuum is imposed to the system and maintained in order to assure leakage absence. Finally, the resin was injected using negative pressure suction. In the case of the thermoplastic resin Elium, the applied mixing ratio is ELIUM 188O 100 wt.% and curing agent BPO (Benzoyl Peroxide) 1.5 wt.%. For the Epoxy resin EPR-L20, the mixing ratio is L-20 100 wt.% and curing agent 33 wt.%. After resin injection, each laminate was cured at room temperature for 24 h. The Epoxy resin EPR-L20 was post-cured at 60 °C for another 15 h.

The specimens were cut to the equipment standard size (squares of 200 × 200 mm), using machining tools, and positioned in the impact test clamping apparatus, as shown in Figure 6.

### 2.2. C-Scan

A C-scan system (YASKAWA MOTOMAN Co. Ltd., Kitakyushu, Fukuoka, Japan, model CAUS-312) was used to verify the damage within the impacted specimens. Images were produced by ultrasonic wave transmission through water. The reflected wave phase and magnitude analysis allowed to visualize the state of the interfaces on the interior of the impacted specimens.

### 2.3. Numerical Finite Element Models

Auxiliary numerical simulations were developed to better understand the test physics and results, covering each drop weight impact test case. The commercial finite element software ABAQUS (v.6.14) explicit solution was selected as the dynamic model simulation platform. Eight different finite element models were developed, considering each material layup sequence for both Elium and Epoxy resins, as previously presented, totaling 30 experimentally studied configurations. The models were subjected to the same impact energy levels measured in the tests. Figure 7 presents the main parameters used in the model development. The initial impact energy input considers the initial velocity of the impactor, equal to the one measured in each case during the experimental tests. Support constraints were applied to the borders of each layer of the composite layups to simulate the clamping apparatus.

The impactor apparatus (25 kg total mass) may be described as a two degrees of freedom (2DOF) system, for both numerical simulation and experimental test data analysis. The system is represented by the dynamic system presented in Figure 8, dividing the total mass between the impactor base mass (M1 = 19.4 kg) and the impactor probe mass (M2 = 5.6 kg). Damping was disregarded, both for the load cell and plate, during the construction of the numerical models.

During the impact tests, forces were measures by a load cell (LC) placed between the two main parts, represented herein by the two masses in the model. In the proposed numerical models, the force measurements are possible to be obtained both at the load cell and at the impactor tip positions. Considering the impactor system as the proposed dynamic model, one can predict a force increment in the values actually applied to the plate if compared to the load cell measurements. Focusing on a direct finite element numerical simulation versus experimental data comparison, a 2DOF impactor model was developed using two masses attached by a contact surface, allowing to integrate forces at both load cell and impactor tip positions. The forces at the load cell position were used to compare numerical finite element results to experimental data in order to correctly simulate the measured load cell signal. Contact model was also applied between the impactor and plies, including self-contact between parts, in order to perform the impact explicit simulation. A parametric study was conducted to verify the estimated stiffness of 90% of steel Young’s modulus, adopted to the load cell model, demonstrating a small influence level (i.e., below method’s accuracy) on the results obtained using the measurement methods proposed in this work. Figure 9 shows normalized forces, integrated in both load cell and impactor tip model positions. Discrete values between 1% and 100% of steel Young’s modulus were considered as reference.

A fixed edge boundary condition was applied to the laminated plate finite element model borders (Figure 10a). Between each layer mesh (Figure 10b), a cohesive zone contact interface [56] was used to model the interfaces, covering the region around the impact point [green area in Figure 10c], to verify the influence of the delamination. Tie constraints were inserted in the remaining regions far from the impact point (gray area in Figure 10c) in order to reduce computational processing costs. The impactor was divided in an impactor base rigid model, and an also rigid impactor probe model (Figure 10d).

The mesh convergence study was developed using model meshes with totals of 19,650, 24,550, 29,694, 47,636, and 91,750 elements for the glass/thermoset case. A low velocity impact explicit case was simulated in each configuration, with sufficient energy to generate damage with no perforation, in order to verify the mesh influence in the results. The area measurement process applied in these studies is described in Section 2.4. During mesh convergence studies CZM (cohesive zone model) and Hashin failure criteria were analyzed separately aiming to better visualize each damage parameter. Figure 11 shows, for each proposed mesh configuration, three damaged area measurements: (i) an intra-laminar damage envelope, considering Hashin criteria only; (ii) an inter-laminar damage (delamination) envelope, considering cohesive criteria only; (iii) an intra-laminar and delamination superimposed damage envelope, considering both Hashin and cohesive damage results. For each defined mesh, the damaged area is calculated applying the three proposed envelope cases, and a mesh convergence parameter is defined comparing each damaged area value with its counterpart obtained using the following refined mesh. The computational time cost to process each case was computed using an Intel^®^ Xeon^®^ CPU E5-1607 (Santa Clara, CA, USA) v2 3.00 GHz machine with 16 GB of installed RAM. A mesh convergence criterion was defined considering the damaged area shapes separately and, based on the calculated areas, a 5% maximum measurement variation was assumed as the maximum accepted difference to next refined mesh. The 47636 elements mesh case was selected, considering the aforementioned criteria, as a tradeoff between computational costs and expected obtained results when applying the proposed mesh to the auxiliary numerical simulations.

Each full finite element model was created in ABAQUS based on the arrangements of unidirectional ply parts, each composed of 6728 nodes and 3249 SC8R linear hexahedral continuum shell elements [57], oriented and arranged according to the respective layup, where thickness was determined from the element nodal geometry and only the displacement degrees of freedom were taken into account. The impactor was modeled with two rigid bodies: the impactor base rigid model, composed by 202 nodes and 110 C3D8R linear hexahedral elements [57], and the impactor probe rigid model including the tip diameter, composed by 466 nodes and 2040 C3D4 linear tetrahedral elements [57]. Thus, the complete numerical models for carbon, hybrid1 and hibrid2 layups (16 plies), for both Elium and Epoxy matrices, were composed by a total of 108,314 nodes and 54,134 elements (51,984 SC8R linear hexahedral, 2040 C3D4 linear tetrahedral, and 110 C3D8R linear hexahedral elements). On the other hand, for glass/Elium and glass/Epoxy (layups with 14 plies), the models were composed by a total of 94,858 nodes and 47,636 elements (45,486 SC8R linear hexahedral, 2040 C3D4 linear tetrahedral, and 110 C3D8R linear hexahedral elements).

#### 2.3.1. Materials

The auxiliary finite element models presented in this work were developed considering literature-based material properties, aiming to better understand the tests’ physics. Table 2 presents the material data for each NCF ply, including the literature data sources. This material data presented a good degree of numerical/experimental comparability when applied to the auxiliary models developed in this work, as well as in other studies in literature [58,59,60,61,62]. Despite that, it is worth emphasizing that detailed studies were not carried out here for numerical or experimental data validation, and more advanced models may be useful to improve numerical results.

An inter-laminar cohesive failure criterion [68] was applied using contact interfaces between each adjacent layer in order to simulate delamination during the impact procedure. Table 3 presents the interface material data, followed by literature data sources. For Hybrid 1 and Hybrid 2 specimens’ cases, the worst case between Elium and Epoxy options were implemented in the models. Stiffness coefficients in the normal and both shear directions, which relates the contact force to the penetration distance, were chosen automatically by ABAQUS/the explicit default contact enforcement method. The applied general contact algorithm uses a penalty method to enforce the contact constraints, such that the effect on the time increment is minimal yet the allowed mesh penetration is not significant in the analyses [57]. The applied interlaminar model (CZM) was assumed to be analogous to an adhesive model between each meso-scale lamina model, considering loading as purely in-plane, and delamination propagation mainly driven by shear in the homogeneous resin-rich region between plies.

#### 2.3.2. Damage Criteria

Hashin damage initiation criterion [74] even with accuracy limitations, is widely applied in industrial application purposes [75] to predict the onset of damage for anisotropic fiber-reinforced materials and was selected for the auxiliary finite element simulations presented in this work. Damage can be initiated without a large amount of plastic deformations. Four distinct failure modes were considered [57]: tensile fiber failure ($\sigma_{11}>0$), compressive fiber failure ($\sigma_{11}<0$), tensile matrix failure ($\sigma_{22}>0$), and compressive matrix failure ($\sigma_{22}<0$). The undamaged material response is implemented using a linearly elastic model for simplicity, and the damage evolution law is based on the energy dissipated during the damage process and linear material softening [76].

Inter-laminar cohesive damage criteria [56] is also applied to the finite element model in order to simulate delamination and debonding damages between plies. Damage onset is implemented using the quadratic stress criterion, initiating damage when a quadratic interaction function, involving the contact stress ratios, reaches a value of one. The cohesive material response was developed based on the Benzeggagh–Kenane fracture criterion [57,77].

### 2.4. Area Measurement Process

A standard damage measurement procedure is proposed to verify the main damaged area in C-scan images, creating a damage outline, as proposed based by Tan et al. [78]. The maximum damage length was also measured to perform a quantitative reference of non-negligible cracks outside the main damage area. An image processing routine was implemented in Python, aiming to reduce the human influence on the presented measurements. In this procedure, the raw image was first imported and converted to a grayscale array. Otsu’s method [79] was applied to calculate an optimal threshold, maximizing the variance between two classes of pixels and minimizing the intra-class variance [80]. Using the obtained threshold as reference, the in-house semi-automatic tool tracks borderline polygons, filters the obtained data, and calculate its area, and the maximum damage length for each studied case. It is important to highlight that C-scan results, besides presenting a damage shape regularly close to the real specimen, are mostly overestimated [81,82]. This procedure makes it easier to visualize image variations observed in the C-scan, mostly already overestimated as mentioned before, leading the analysis to the safety side, but care must be taken not to over-reinforce the final structure.

The semi-automatic tool aims to achieve a fair judgement on the measurement process in experimental results, also being adapted to measure the main damage area in auxiliary finite element model graphical results. Focusing on obtaining comparable results to C-scan images, as proposed by Liu at al. [75], superimposed model images from each ply in the numerical results were used to perform a delamination envelope and an intra-laminar damage envelope. The image-to-binary process for Hashin and cohesive damage results are processed separately, and a Boolean operation is used to superimpose both images. Then the routine proceeds finding damage polygon and calculating its area.

## 3. Results

### 3.1. Experimental Studies

This section presents the obtained results in the proposed tests. Force measurements obtained during the experiments for each case are presented in Figure 12, as impact force vs. time plots for different energy levels and resin systems. It is possible to observe that these curves are not smooth for carbon fiber specimens due to penetration of the impactor in the panels. Additionally, it is possible to observe that penetration acts as a limitation of maximum measured impact forces in carbon specimens.

Figure 13 presents experimental results for carbon fiber with Elium and Epoxy resin systems. The columns depict the impact energy level for each resin system. The first two rows present front and back images from the impacted specimens, respectively. The obtained C-scan images are shown in the third, fourth and last row and contain images of the damage measurements, as previously explained. This presentation scheme is also used in the following figures for different fiber systems test results. In Figure 13 it is possible to observe a larger damage in Epoxy specimens as compared to Elium specimens, for both main damage and maximum length measurements. Considering the 97 J impact case, an almost complete perforation was observed for both matrices. Notice that due to the fragile failure characteristics of the carbon/Epoxy, very pronounced cracks may be seen if compared to its carbon/Elium counterparts. Comparing the measured areas for both matrices at each energy level, it is possible to observe a trend of increasing damage when conventional epoxy resin is applied: from 787 mm^2^ to 1136 mm^2^ for 57 J energy level, and from 1552 mm^2^ to 1624 mm^2^ for 97 J energy level. No impact tests were performed for epoxy specimens at 83 J for all the studied layups.

For the glass fiber cases, as presented in Figure 14, very small damage levels were observed for the tested energy levels, even though, as for carbon iber, larger damages are observed for Epoxy specimens when compared with Elium counterparts: an increment from 211 mm^2^ to 251 mm^2^ for 57 J energy level, and from 257 to 349 mm^2^ for 97 J energy level may be observed.

Figure 15 presents Hybrid 1 carbon/glass layup test results, where damage can be visually observed in the back images (second row), due to the contrast of the delaminated bottom glass layer. Again, it is possible to observe a damage level increment in the Epoxy specimens, comparing with Elium specimens: from 422 to 528 mm^2^ for 57 J energy level, and from 906 to 1012 mm^2^ for 97 J energy level. Considering the maximum length measurement parameter, a slight increment may be observed for 97 J (55 to 58 mm) and no increment was observed for 57 J case. Though no impact tests were performed for epoxy specimens at 83 J, it is possible to observe that the maximum length measurement parameter for Elium achieved the same value as the one obtained for Epoxy at 97 J.

The Hybrid 2 carbon/glass layup test results are presented in Figure 16. Similar to previous results, visual damages can be easily observed in the front images (first row) due to the contrast of the delaminated top glass layer. It is possible to verify the trend of larger damage levels, considering both measurement parameters, for Epoxy specimens comparing with Elium specimens: from 370 to 499 mm^2^ for 57 J energy level, and from 723 to 906 mm^2^ for 97 J energy level, and, considering the maximum length measurement parameter, an increment may be observed for 97 J (47 to 55 mm) and for 57 J (27 to 54 mm) cases.

As observed in the literature [6,7,32,33,37,83] regarding failure modes, Elium-based specimens, if compared to Epoxy-based counterparts, presents a less brittle failure (fracture) response due to the material plastic behavior. Additionally, it is important to observe that glass-fiber material based specimens are typically far less brittle than its carbon reinforced counterparts [84], which leads to the observed advantages in glass built specimens where, besides the carbon stiffness and strength intrinsic advantages. Hybridization, jointly with Elium thermoplastic resin application, is observed as a promising possibility to improve materials for advanced composite design.

### 3.2. Numerical Studies

Based on finite element analyses simulations, Figure 17 presents results for carbon fiber specimens, observing a fairly good damage level correlation compared to experimental results (Figure 13), even though using a simplified numerical model. As in the figures for experimental results, the columns present different impact energy levels in simulations for each resin system. The first row presents a comparative strain distribution between results obtained for each studied case. To directly compare the damage levels to those obtained for C-scan results of tested panels, the second row presents the main damage area measured through the superimposed envelope considering both failure criteria, as previously described. These damage envelopes results were obtained considering superimposed results for each ply using the Hashin failure criterion [74] in a Boolean operation with superimposed results for each interface damage using the cohesive criterion [56]. This presentation scheme is followed in the sequent figures for simulation results of different fiber systems. One can observe in Figure 17, considering the top row, the presence of concentrated stresses outside the main impact region. In this region, in the lower row, it is possible to observe cracks in the specimens, also observed in the experimental results. It is possible to observe a damage level increment in the Epoxy specimens, comparing with Elium specimens: from 718 to 821 mm^2^ for 57 J energy level, from 1039 to 1117 mm^2^ for 83 J energy level, and from 1063 to 1207 mm^2^ for 97 J energy level.

Glass finite element simulation results are presented in Figure 18. It is possible to observe considerably larger damage levels in those numerical results, for both resins, when compared with those present in experimentally tested specimens (Figure 14). As well as that, it is possible to observe a damage level increment in the Epoxy specimens, comparing with Elium specimens: from 79 to 377 mm^2^ for 57 J energy level, from 517 to 626 mm^2^ for 83 J energy level, and from 616 to 725 mm^2^ for 97 J energy level. No significant cracks were noticed outside the main impact region, considering both strain distribution and damage envelope, agreeing to the experimental results.

Figure 19 presents finite element simulation results for the Hybrid 1 layup cases. A fairly good correlation can be verified of damage levels compared to experimental results (Figure 15), even though a simplified numerical model was used. It is also possible to observe a damage level increment in the Epoxy specimens, comparing with Elium counterparts: from 444 to 547 mm^2^ for 57 J energy level, from 699 to 826 mm^2^ for 83 J energy level, and from 903 to 988 mm^2^ for 97 J energy level. Small cracks were noticed outside the main impact region, considering both strain concentrations and damage envelope, agreeing to experimental results.

Results for finite element simulation for Hybrid 2 layup cases are presented in Figure 20. Again, it is possible to observe a reasonable numerical versus experimental correlation for damage levels considering experimental results (Figure 16). Again, it is also possible to observe a damage level increment in the Epoxy specimens, comparing with Elium counterparts: from 493 to 562 mm^2^ for 57 J energy level, from 680 to 737 mm^2^ for 83 J energy level, and from 880 to 902 mm^2^ for 97 J energy level. Small cracks were noticed outside the main impact region, considering both strain concentrations and damage envelope, agreeing to experimental results.

### 3.3. Analysis

Considering both the experimental and numerical results which Figure 21 presents, for each studied case, the largest load cell force measurement during the impact process. These results were considered as fairly adequate for the relatively simplified finite element deterministic models. 

The damaged area and maximum damage length results are condensed Figure 22. Although carbon/Elium properties were approximated using carbon/Epoxy stress and strength data, finite element models present similar patterns to experimental data for both carbon and hybrid cases. Additionally, due to the model limitations, this simplified numerical approach may be non-conservative, as verified in some Hybrid 1 cases, serving only as a reference for studies. It is also possible to verify that carbon/Epoxy numerical results show smaller secondary intra-laminar damage (maximum length), in addition to the main measured damage, if compared to the carbon/Elium counterparts.

Considering the tested energy levels, for both Elium and Epoxy cases, small damages are observed for glass specimens in Figure 22. It is also possible to observe that the corresponding finite element results present a trend to simulate larger damage levels as compared with experimental results.

Also, according to Figure 22, small variation was observed between the test damage levels for carbon/Elium and carbon/Epoxy at 97 J impact energy level, considering the damaged area measurement procedure. However, the maximum damage length “secondary failures” are quite considerable for the carbon/Epoxy experimental case. Carbon/Epoxy’s relatively small main damage area may be explained by its fragile failure characteristics, since damage tends to be more localized, similarly to the expected result in a high energy ballistic test [18,70].

Hybrid/Elium panels show less impact damage than hybrid/Epoxy panels, again corroborating with the literature results [7,32,33,53,66,70] for thermoplastic versus thermoset resins with different reinforcement materials (Figure 22).

## 4. Discussion and Conclusions

An experimental impact test program was conducted in composite panels using a thermoplastic and thermoset resin systems reinforced by carbon, glass and hybrid car-bon/glass fibers layups. Only one impact test was performed for each configuration at each energy level.

As well as the similarities between Elium and a conventional thermoset matrix, in our results Elium (thermoplastic) panels presented less impact damage levels than Epoxy (thermoset) panels for the proposed layups (glass, carbon and hybrids). Overall, considering the results and the proposed measurement methodology, thermoplastic resin (Elium) layups, compared to its thermoset counterparts, presented reduction of damage areas for carbon, glass and hybrid fiber systems. Despite the lack of statistical representation, given that only one test was performed for each studied point, a trend of impact damage area reduction for Elium specimens was observed when considering all the studied cases. The studied Elium-based specimens presented in our studies a less brittle failure (fracture) response compared to Epoxy-based counterparts.

Glass-fiber material-based specimens presented, as typically expected, a far less brittle behavior if compared to its carbon reinforced counterparts. As well as carbon stiffness and strength intrinsic advantages, for both Elium (thermoplastic) and Epoxy (thermoset), glass showed the lowest damage levels while carbon showed the highest. It was observed that the presence of glass layers decreases the damaged area in both hybrid layup cases, since glass is more energy absorbent than carbon. The use of hybrid layups demonstrated to be a way to achieve a compromise solution between impact damage resistance and stress/strength characteristics in engineering applications. This should be added to the benefits of impact damage area reduction effect presented due to use of Elium thermoplastic resin.

Due to the inherent difficulties of a visual inspection of composite material, one may observe that a larger damaged area is verified considering the C-scan. On the other hand, it is important to highlight that C-scan tends to lead to overestimated results, besides presenting a damage shape regularly close to the real specimen’s damage. A measurement procedure was proposed herein to easier visualize damage variations in images. The tendency of obtaining overestimated results in C-scan exemplifies the difficulty of identifying composite material impact failures, even with the aid of specialized equipment.

Numerical models are commonly applied to assist and reduce testing costs. The auxiliary numerical models applied in this work, besides its simplicity, presented similar patterns if compared with experimental results, but caution needs to be taken since, as shown, some cases presented non-conservative results when comparing to experimental data. Even though it is possible to observe a considerable difference, which might be reduced using a higher fidelity finite element model for impact simulation, the relatively low fidelity models presented are useful to understand the physics and the results of the proposed problem.

Although carbon fiber composites are lighter and stronger, wind turbines blades are commonly built of glass fiber composites. Hybrid carbon/glass fiber composites became a feasible alternative to balance cost-performance for larger wind turbine blades. Applying Elium thermoplastic resin, jointly with hybridization, can be seen as a promising possibility to improve advanced composite materials design. Elium resin demonstrates to be competitive to replace traditional thermoset options, with advantages related to better reuse capabilities, ease of fabrication, and good impact and damage tolerance properties.

## Figures and Tables

**Figure 1 materials-14-06377-f001:**
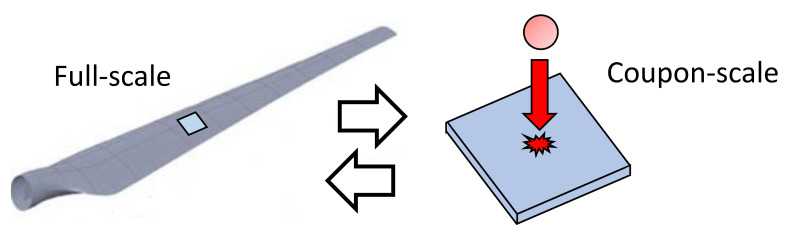
Scale of investigation for impact in wind turbine blades.

**Figure 2 materials-14-06377-f002:**
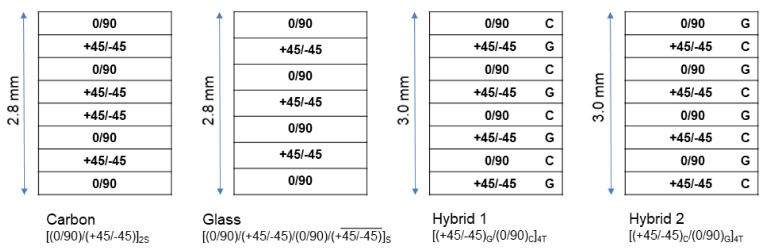
Lay-up sequence of carbon, glass and hybrid plates.

**Figure 3 materials-14-06377-f003:**
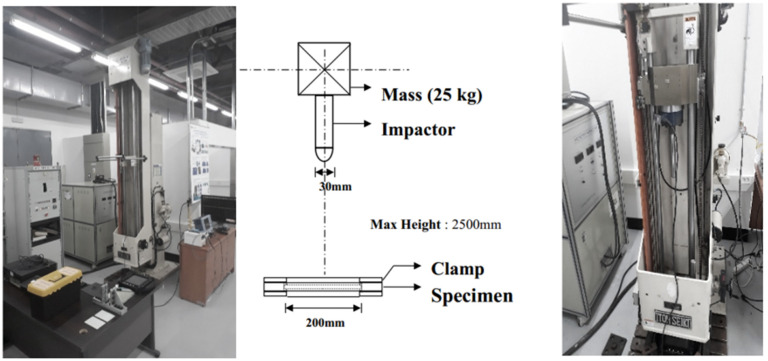
Impact drop tester machine specification (ITOH SEIKI Co. Ltd., Chonburi, Thailand).

**Figure 4 materials-14-06377-f004:**
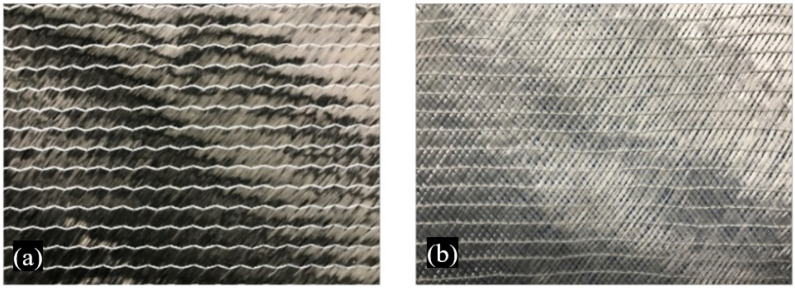
BX45° NCF fabrics: (**a**) carbon (Chomarat Co., Ltd., Le Cheylard, France) and (**b**) glass (HANKUK Carbon Co., Ltd., Seoul, Korea).

**Figure 5 materials-14-06377-f005:**
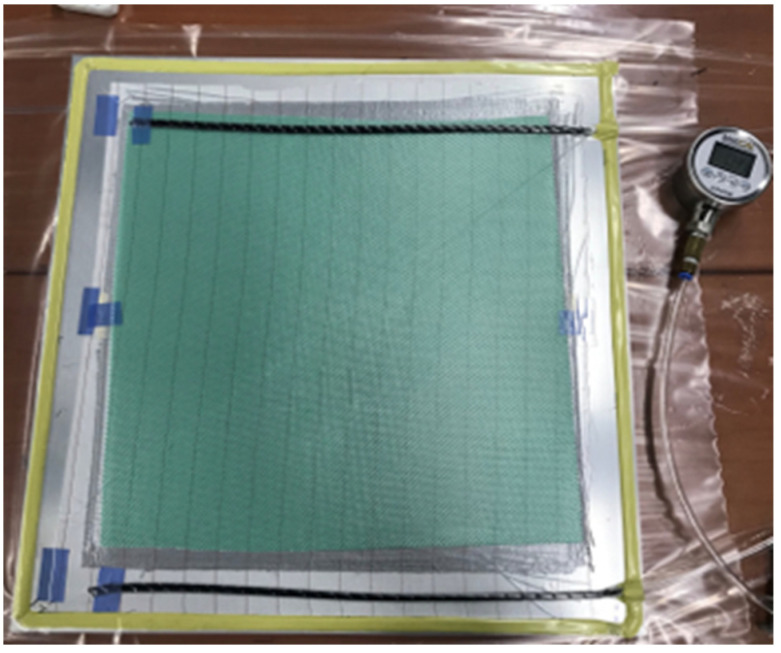
Specimen manufacturing process.

**Figure 6 materials-14-06377-f006:**
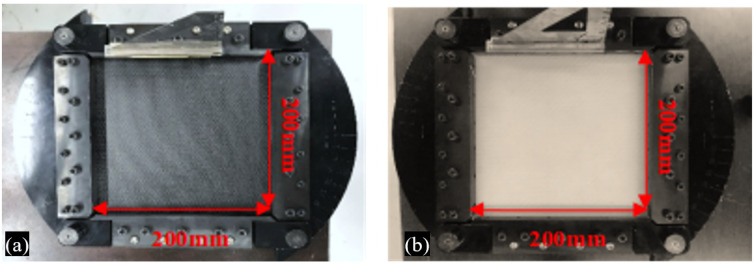
Specimens in the clamping apparatus: (**a**) carbon specimen and (**b**) glass specimen.

**Figure 7 materials-14-06377-f007:**
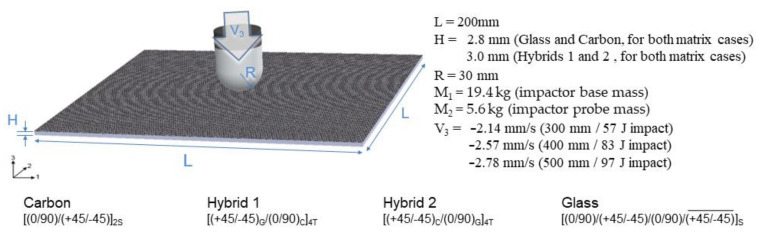
Model parameters.

**Figure 8 materials-14-06377-f008:**
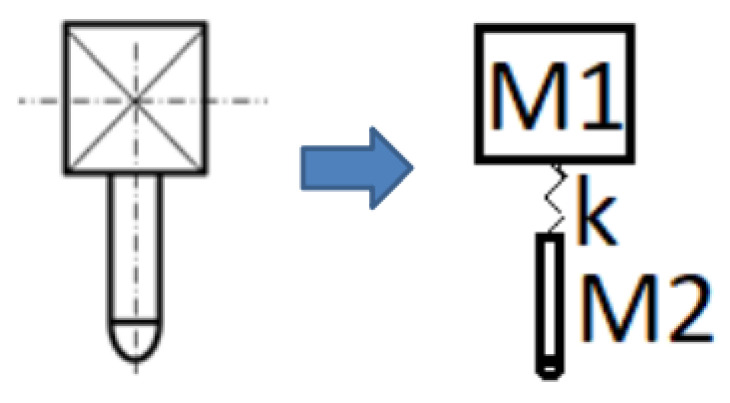
Proposed 2DOF model for the impactor system: simplified finite element model data representation.

**Figure 9 materials-14-06377-f009:**
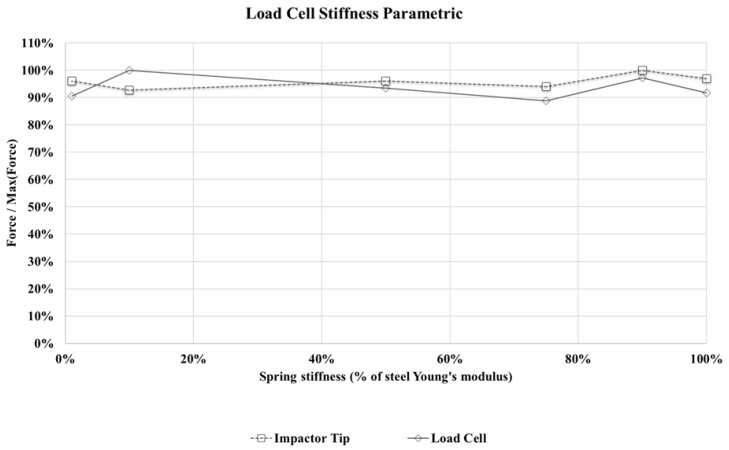
Force measurements parametric studies in both load cell and impactor model positions, normalized by maximum measurement for each case.

**Figure 10 materials-14-06377-f010:**
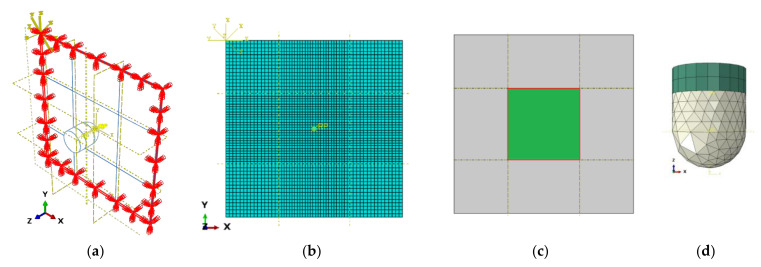
Model (**a**) boundary conditions, (**b**) single ply mesh, (**c**) inter-ply tie constraints and cohesive zone contact interface regions, and (**d**) two degree of freedom impactor mesh.

**Figure 11 materials-14-06377-f011:**
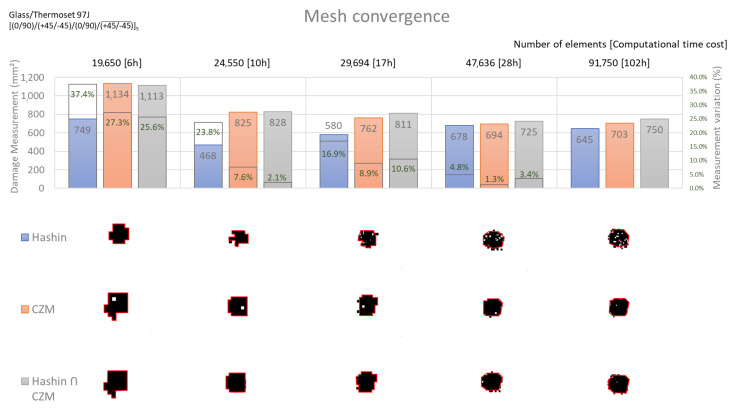
Mesh convergence analysis.

**Figure 12 materials-14-06377-f012:**
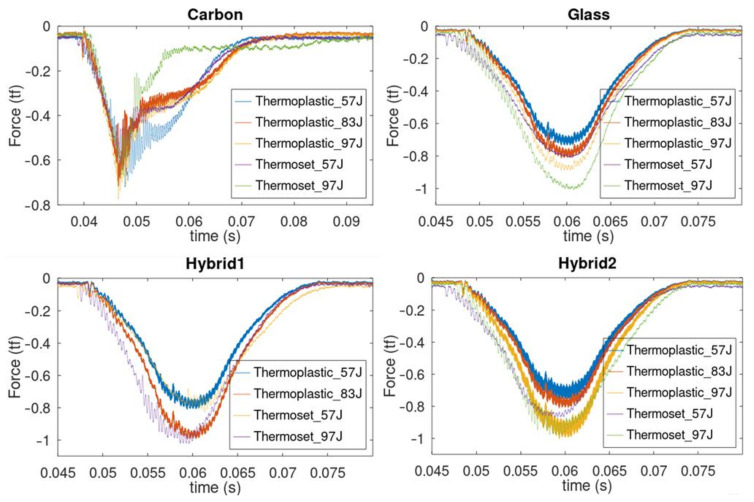
Experimental impact forces measured for Elium (thermoplastic) and Epoxy (thermoset) at different energy levels.

**Figure 13 materials-14-06377-f013:**
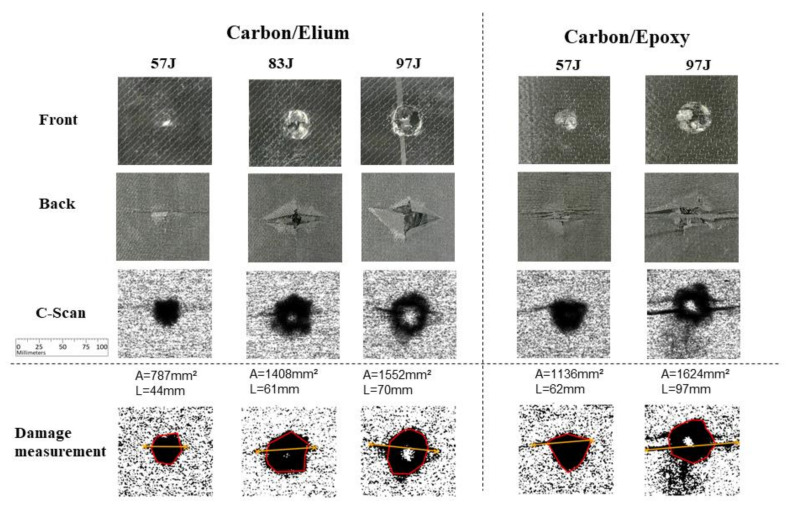
Experimental results for Carbon fiber specimens.

**Figure 14 materials-14-06377-f014:**
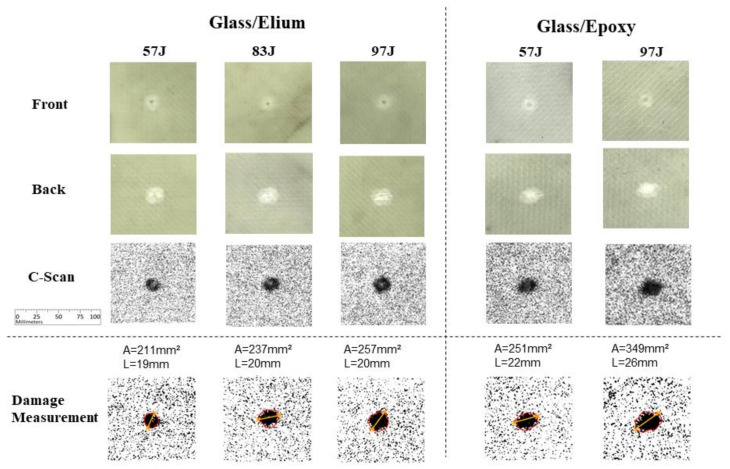
Experimental results for glass fiber specimens.

**Figure 15 materials-14-06377-f015:**
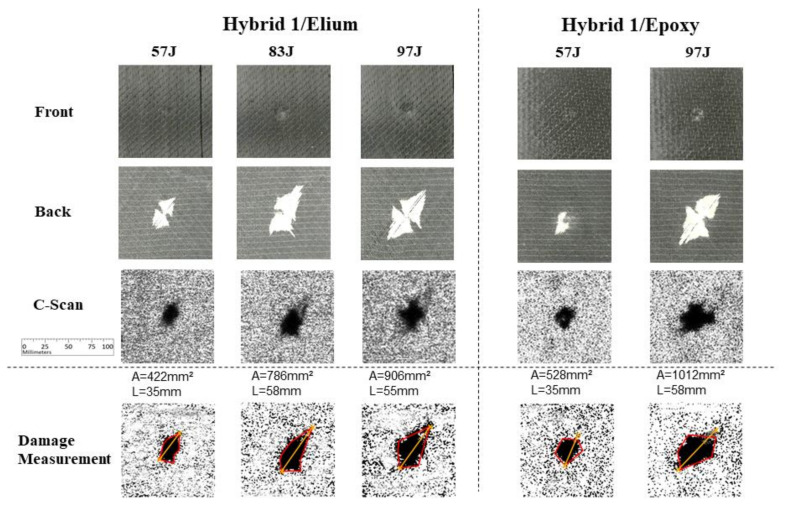
Experimental results for Hybrid 1 carbon/glass fibers specimens.

**Figure 16 materials-14-06377-f016:**
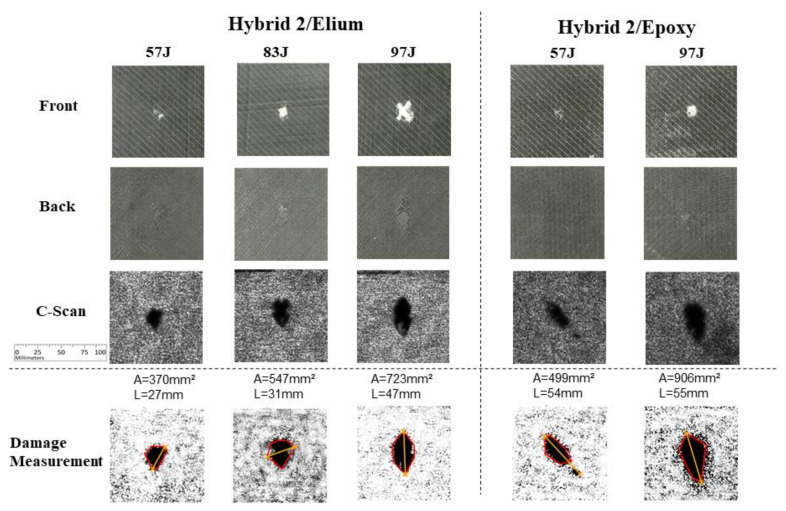
Experimental results for Hybrid 2 carbon/glass fibers specimens.

**Figure 17 materials-14-06377-f017:**
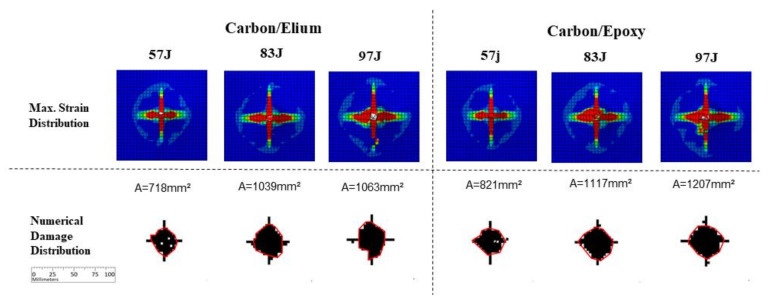
Numerical results for carbon fiber specimens.

**Figure 18 materials-14-06377-f018:**
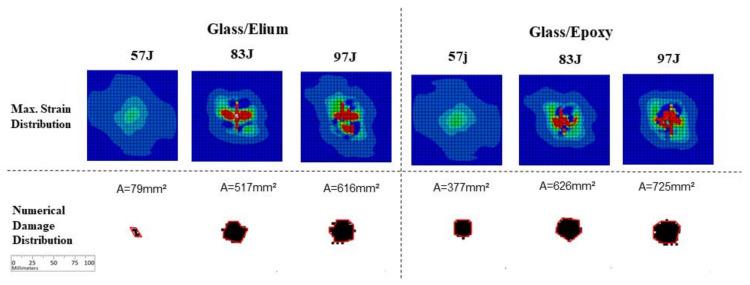
Numerical results for glass fiber specimens.

**Figure 19 materials-14-06377-f019:**
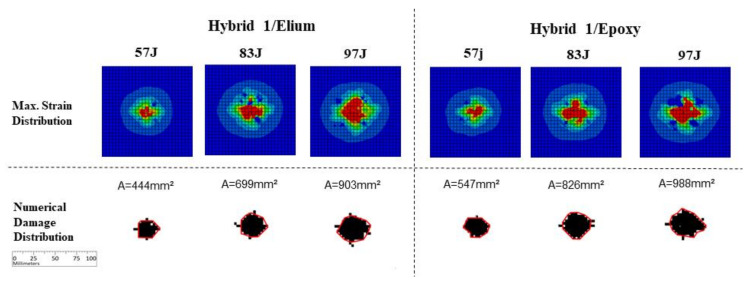
Numerical results for Hybrid 1 carbon/glass fibers specimens.

**Figure 20 materials-14-06377-f020:**
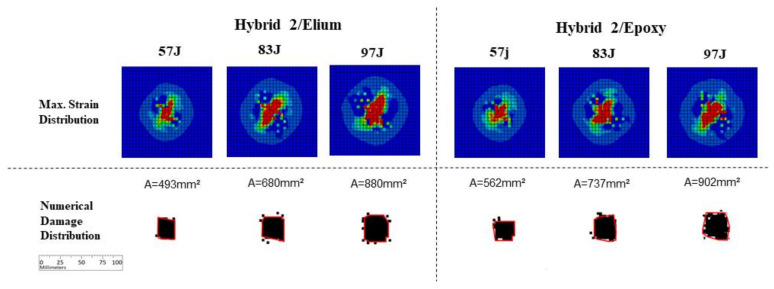
Numerical results for Hybrid 2 carbon/glass fibers specimens.

**Figure 21 materials-14-06377-f021:**
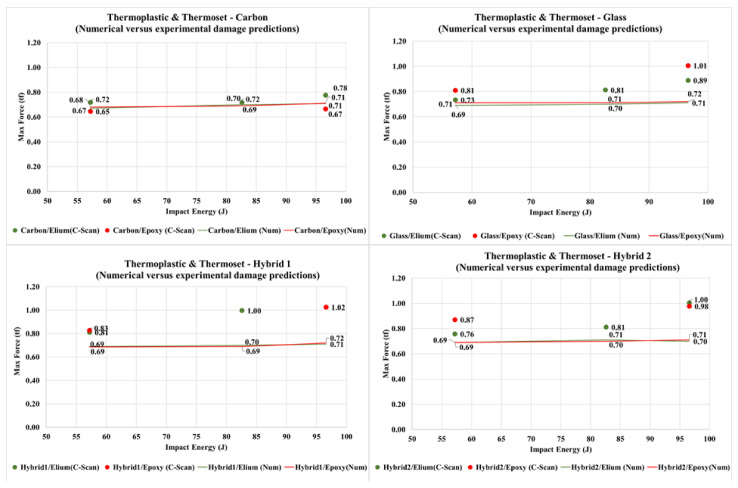
Numerical vs. experimental maximum force measurements as function of impact energy levels for different fiber and resin systems.

**Figure 22 materials-14-06377-f022:**
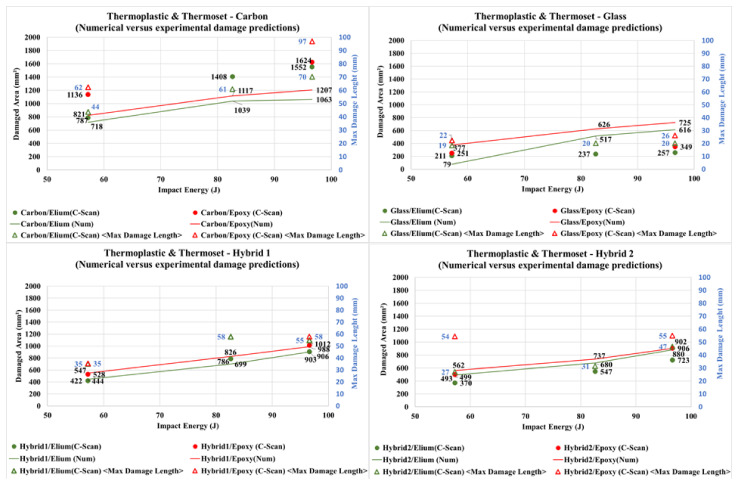
Numerical versus experimental damage predictions.

**Table 1 materials-14-06377-t001:** Proposed analysis cases.

	Reinforcement	Matrix	Layup Sequence
Carbon/ELIUM	Carbon NCF (Chomarat Co., Ltd., Paris, Paris Region France)	ELIUM 188O (Arkema Co., Ltd., Lacq, Pyrénées-Atlantiques, France)	[(0/90)/(+45/−45)] _2S_
Carbon/Epoxy		EPR-L20 (Hexion Co., Ltd., Colombus, OH, USA)	
Glass/ELIUM	Glass NCF (HANKUK Carbon Co., Ltd., Miryang-si, Gyeongsangnam-do, South Korea)	ELIUM 188O (Arkema Co., Ltd., Lacq, Pyrénées-Atlantiques, France)	[(0/90)/(+45/−45)/(0/90)/$\bar{\left(+45/-45\right)}$] _S_
Glass/Epoxy		EPR-L20 (Hexion Co., Ltd., Colombus, OH, USA)	
Hybrid 1/ELIUM	Carbon NCF (Chomarat Co., Ltd., Paris, Paris Region France) Interleaved with Glass NCF (HANKUK Carbon Co., Ltd., Miryang-si, Gyeongsangnam-do, South Korea)	ELIUM 188O (Arkema Co., Ltd., Lacq, Pyrénées-Atlantiques, France)	[(+45/−45) _G_/(0/90) _C_] _4T_
Hybrid 1/Epoxy		EPR-L20 (Hexion Co., Ltd., Colombus, OH, USA)	
Hybrid 2/ELIUM		ELIUM 188O (Arkema Co., Ltd., Lacq, Pyrénées-Atlantiques, France)	[(+45/−45) _C_/(0/90) _G_] _4T_
Hybrid 2/Epoxy		EPR-L20 (Hexion Co., Ltd., Colombus, OH, USA)	

**Table 2 materials-14-06377-t002:** NCF ply properties used in the analyses.

	Carbon/Elium		Carbon/Epoxy		Glass/Elium		Glass/Epoxy	
Longitudinal Stiffness (GPa)	141	[63,64]	141	[63]	45,0	[64,65]	45	[65]
Transverse Stiffness (GPa)	7.3	[39,58]	7.2	[58]	14.3	[39,66]	14.1	[66]
Shear Stiffness (GPa)	3.4	[58,64]	3.4	[58]	4.9	[39]	6.4	[66]
In-plane Poisson’s Ratio ν12/ν23	0.34	[58,64]	0.34	[58]	0.22	[39]	0.245	[66]
Longitudinal Tensile Strength (MPa)	2040	[63,64]	2040	[63]	1725	[64,65]	1725	[65]
Longitudinal Compressive Strength (MPa)	1192	[58,64]	1192	[58]	620	[64,65]	620	[67]
Transverse Tensile Strength (MPa)	24.8	[58,64]	19.6	[58]	102.1	[39,58,61]	80,6	[58,61]
Transverse Compressive Strength (MPa)	94.7	[58,64]	92.3	[58]	330.3	[39,58]	322	[58]
Shear Strength (MPa)	51	[58,64]	51	[58]	49.7	[66]	54.5	[58]
Longitudinal Tensile Fracture Energy (N/m)	48,400	[58,64]	48,400	[58]	32,000	[58,64]	32,000	[58]
Longitudinal Compressive Fracture Energy (N/m)	60,300	[58,64]	60,300	[58]	20,000	[58,64]	20,000	[58]
Transverse Tensile Fracture Energy (N/m)	4500	[58,64]	4500	[58]	4500	[58,64]	4500	[58]
Transverse Compressive Fracture Energy (N/m)	8500	[58,64]	8500	[58]	4500	[58,64]	4500	[58]

**Table 3 materials-14-06377-t003:** Cohesive contact zone interface properties used in the analyses.

	Carbon/Elium		Carbon/Epoxy		Glass/Elium		Glass/Epoxy	
Maximum Nominal Stress N (MPa)	40	[64,69]	40	[69]	70	[70]	65	[70]
Maximum Nominal Stress S1 (MPa)	50	[64,69]	50	[69]	80	[70]	72	[70]
Maximum Nominal Stress S2 (MPa)	50	[64,69]	50	[69]	80	[70]	72	[70]
Damage Evolution Normal Fracture Energy (N/m)	1970	[71]	293	[69]	1880	[70]	1075	[72]
Damage Evolution 1st Shear Fracture Energy (N/m)	631	[64,69]	631	[69]	3840	[70]	4000	[73]
Damage Evolution 2nd Shear Fracture Energy (N/m)	631	[64,69]	631	[69]	3840	[70]	4000	[73]
Benzeggagh-Kenane Exponent	1	[64,69]	1	[69]	1	^a^	1	^a^

^a^ Approximated using same exponent as carbon counterpart.

## Data Availability

Data is available upon request from the corresponding author.

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
