# Peer review of "Experimental and Numerical Comparison of Impact Behavior between Thermoplastic and Thermoset Composite for Wind Turbine Blades"

_materials, 2021, doi:10.3390/ma14216377_

Round 1
Reviewer 1 Report
The authors present their results concerning the (low velocity) impact behavior on thermoplastic and thermoset composites with carbon, glass and 2 carbon/glass hybrid fibers in the NCF (Non-Crimp Fabric) architecture. Experimental data is compared to simulations. This is a nice and correct study, based on a good design, allowing pertinent conclusions. Could be published in actual form.
Reviewer 2 Report
At the outset I congratulate the authors for this work. The role of impact loading on composites is of importance to understand as such composites may become vulnerable. In that context, I consider that the article is topical (with appropriate timing) and interesting, and hence I recommend for publication. A few questions:
- How were the lay-up procedures shown in Fig. 2 developed/selected?
-
The details on the FE models are quite light. Please ensure that your model is properly described to enable interested researchers from extending and replicating your work. Special attention should be paid to specifics such as element type (DOFs), CZM inputs, convergence criteria and performance metrics.
- Between Fig. 18-20, please indicate which strain distribution is more favorable.
Reviewer 3 Report
Review Materials 1338314
Title Experimental and Numerical Comparison of Impact Behavior 2 Between Thermoplastic and Thermoset Composite for Wind 3 Turbine Blades.
This paper studies experimentally and numerically the impact behavior of composite materials for wind turbine blades. Non-crimp fabrics of carbon fibers and glass fibers were combined with two different matrices, one thermoplastic and one thermoset. The main goal is to compare the composite system with different matrices under a low-velocity impact event. Monolithic systems and hybrid systems were fabricated via vacuum-assisted resin transfer mold. The damage due to the impact event has been evaluated experimentally and numerically showing slightly better damage resistance in the systems fabricated with thermoplastics.
The paper is well organized, but there are sections over detailed and others under detailed. The English must be reviewed with a grammar corrector. In addition, the discussion and conclusion must be more realistic. The numerical differences showed in the results are small, and general conclusions must be handle with care.
The paper can be a good candidate for the proposed journal, but major issues must be corrected to reach a good scientific level.
The reviewer has the following major observations:
- The glass laminate is not quasi-isotropic. I understand that the authors want to keep a thickness to compare different systems. Please, explain the main goal of the experiments indicating that the glass laminates are not complete quasi-isotropic but it fits in the goal of the experimental device. The authors must clarify if the laminates are symmetric or not. Figure 2 shows non-symmetric laminates because there is no mirroring at the symmetric plane. This must be also indicated in the text, and evaluated the relevance.
- Too many details in the manufacturing process with figure 5. I think the most important features are the differences between the two matrix systems. If there are some specific details in the fabrication between them, in addition to the text I will include the figures. If not, I will indicate the final results of both systems, epoxy versus thermoplastic
- In addition to the previous point. It is relevant the C-Scan of the manufactured plates if there are important differences.
- I understand the difficulties of modeling the impactor. It is important to demonstrate with the outputs of the simulations that the pressure evaluated in the simulated load cell is working properly because there is a stiffness of the material that can affect the result. Please, check that the stiffness used does not affect the numerical results. In addition, the local force calculated at the tip of the impactor is not a usual measurement. If this value is going to be reported, the value must be included in the convergence study where the refinement of the impactor is critical. Figure 23, shows that impactor force is a local value. Please, reconsider deleting this numerical measurement or justify it correctly.
- Be careful with the mechanical properties of glass epoxy and thermoplastic. The assumed transverse tensile strength is large compare with usual composites. References do not show the value is experimental or is a fitting value for non-crimp fabrics. This must be detailed in the manuscript, and justify why values are larger than usual.
- Table 3. The shear fracture energy of glass epoxy and thermoplastic must be also justified. The references indicated are studies about adhesives. Why have you selected these values?
- Figure 11, and figure 12 are well-known theoretical models for damage simulation. The authors are over-explained the damage model without the intention of modifying it or improve it in the study. The explanation must be simplified, indicating the important point of the damage behavior, intralaminar or interlaminar damage, and indicate the correct references.
- On the other hand, the area measurements process is important post-processing customized by the authors for the experiments and simulations of the experiments. Please indicate the main advantage of the post-processing and if possible a quantitative error.
- Results about C-Scans and simulations are compared without quantitative values. The values come later in figure 24. These values are relevant, they must be included from the beginning in the discussion. From the images, it is not clear that thermoplastics have better damage resistance.
- Discussion and conclusion are weak. This is the major issue of the work. Please, review the data, simulations, short them considering the values and errors in the measurements. The authors must be critical of their results. Is there a quantitative difference in damage resistance (damaged area) between the different studied systems? Note that the thermoplastic selected behaves as a thermoset. I think the authors have the data and the knowledge to clarify the results of the present study. I have assumed that in the experimental campaign several experiments with the same material and energy were done. If only one experiment per material and energy has been done, the focus of the study must be completely different. No general conclusion can be done from limited experiments.
And observations through the text:
- Line 17. Please, review the manuscript with a grammar corrector. For example, the preposition and article must be added in line 17. “focusing on the design”.
- Line 24. Please clarify the message “to be also compared to experimental data”
- Line 69. “Full recyclability” Thermoplastics are far from full recyclability. Although they can be modified at high temperatures, they cannot be recycled (melt and use again for the same application). They can be easily reused (better than thermosets) but recycling is a big issue with thermoplastics and thermosets. Please, reformulate the sentence and the indication in the paragraph about full recyclability, and distinguish between recycling and reuse.
- Line 73, thermoplastics suffer chemical changes when heated as thermosets. Correct the sentence with suitable expression chemically correct.
- Line 78 “full material recyclability” Not really true.
- Line 82. Thermosets show also viscoelastic behavior. I am almost sure that the thermoplastic used in the present work shows the same viscoelasticity as another epoxy matrix. The sentence must be clarified, the general behavior must be corrected.
- Line 277. Correct the grammar expression “Damage can be initiated without large amount”
- Line 348. Correct figure 14, not 18.
- Figure 13, the legends of subfigures have a small font. Please, increase them.
- Figure 14, Carbon epoxy 97 J. There is a horizontal crack. The crack does not mention in the text. Is this the perforation case? Please clarify.
- Figure 14 has also small fonts for some text. They must be increased to be readable. This also happens in figure 15, 16, 17, 19, 20 and 21
- Line 367. “Larger damage levels” This cannot be seen in the figure, neither in the values of the area. Please, indicate the values that confirm this fact.
- Line 403. The sentence must be rewritten to avoid confusion. Glass simulations show a larger damaged area, but the contours considering the methodology are not reflected, so the comparison cannot be done a priori.
- Figure 24. Fonts are small. Error data for each experiment must be indicated. If only one experiment per energy has been done this must be indicated and justified. A limited conclusion can be done.
